# Randomised, controlled, open label, multicentre clinical trial to explore safety and efficacy of hyperbaric oxygen for preventing ICU admission, morbidity and mortality in adult patients with COVID-19

Anders Kjellberg [1,2] Johan Douglas,[3] Michael T Pawlik,[4] Michael Kraus,[5] Nicklas Oscarsson [6] Xiaowei Zheng [7] Peter Bergman [8] Oskar Frånberg [9] Jan H Kowalski,[10] Sven Paul Nyren,[7,11] Mårten Silvanius [12,13] Magnus Skold,[14,15] Sergiu-Bogdan Catrina [7,16] Kenny A Rodriguez-Wallberg [17,18] Peter Lindholm [1,19]

KAR-W and PL are joint senior authors.

For numbered affiliations see end of article.

**Correspondence to**
Anders Kjellberg;
anders.kjellberg@ki.se

## ABSTRACT

**Introduction** COVID-19 may cause severe pneumonitis and trigger a massive inflammatory response that requires ventilatory support. The intensive care unit (ICU)-mortality has been reported to be as high as 62%. Dexamethasone is the only of all anti-inflammatory drugs that have been tested to date that has shown a positive effect on mortality. We aim to explore if treatment with hyperbaric oxygen (HBO) is safe and effective for patients with severe COVID-19. Our hypothesis is that HBO can prevent ICU admission, morbidity and mortality by attenuating the inflammatory response. The primary objective is to evaluate if HBO reduces the number of ICU admissions compared with best practice treatment for COVID-19, main secondary objectives are to evaluate if HBO reduces the load on ICU resources, morbidity and mortality and to evaluate if HBO mitigates the inflammatory reaction in COVID-19.

**Methods and analysis** A randomised, controlled, phase II, open label, multicentre trial. 200 subjects with severe COVID-19 and at least two risk factors for mortality will be included. Baseline clinical data and blood samples will be collected before randomisation and repeated daily for 7 days, at days 14 and 30. Subjects will be randomised with a computer-based system to HBO, maximum five times during the first 7 days plus best practice treatment or only best practice treatment. The primary endpoint, ICU admission, is defined by criteria for selection for ICU. We will evaluate if HBO mitigates the inflammatory reaction in COVID-19 using molecular analyses. All parameters are recorded in an electronic case report form. An independent Data Safety Monitoring Board will review the safety parameters.

**Ethics and dissemination** The trial is approved by The National Institutional Review Board in Sweden (2020-01705) and the Swedish Medical Product Agency (5.1-2020-36673). Positive, negative and any inconclusive

### Strengths and limitations of this study

► Randomised controlled clinical trial in compliance with Good Clinical Practice.
► Safety and efficacy endpoints together with multiple explanatory endpoints.
► Independent Data Safety Monitoring Board.
► No placebo, open label.
► Power calculation is based on early pandemic data and 'best practice treatment' have changed during the course of the trial.

results will be published in peer-reviewed scientific journals with open access.

**Trial registration** NCT04327505. EudraCT number: 2020-001349-37.

## INTRODUCTION

### Clinical manifestations and challenges with COVID-19

SARS-CoV-2 was first identified in China in December 2019.[1] The clinical infectious disease COVID-19 was declared a pandemic by the WHO on 11 March 2020; with more than 46 million confirmed cases and more than 1 million confirmed deaths by 2 November 2020.[2] Clinical experience from China and Italy was published early and even though the overall mortality is low (3.4%), the numbers from critical care were fearsome.[3–6] Mortality rates were as high as 90% in patients developing acute respiratory distress syndrome (ARDS) in early reports from Wuhan province. Later reports showed

28-day mortality rates of 61.5% in intensive care unit (ICU) patients with acute respiratory illness.[4] In a recent retrospective cohort study form Wuhan, 19% of patients needed mechanical ventilation or extra corporal mechanical oxygenation, 26% was admitted to ICU and hospital mortality rate was 28%.[7] SARS-CoV-2 enters human cells through ACE2 receptors, abundant in lungs, arteries, heart, kidney and intestines, causing a downstream activation of an inflammatory cascade that activates the innate immune system.[8] A synchronised immune response is vital in the control and resolution of viral infections. In some patients, this activation and resolution is dysregulated, causing a disproportionate reaction, popularly called a cytokine storm.[9] Acute lung injury (ALI) associated with COVID-19 differs from other described ARDS with rapidly progressing respiratory failure and fibrosis. Even patients who have mild symptoms and survive COVID-19 may have significant changes on pulmonary CT, with diffuse ground glass opacities and crazy-paving pattern and consolidation suggesting severe inflammatory involvement.[10] Despite enormous efforts, a definite cure seems far away and there is urgent need for effective treatments to reduce morbidity and mortality. Remdesivir, Hydroxychloroquine, Lopinavir and Interferon-β1a have been tested in a total of 11 266 subjects included, none of the drugs have been proven effective according to recently published results from the WHO Solidarity trial.[11] Corticosteroids were tried early in the pandemic with discouraging results, but recently preliminary results from the RECOVERY-trial showed some reduction in 28-day mortality with dexamethasone, with better effect in severe disease.[12] The RECOVERY-trial showed a mortality of 41.4% in the control group versus 29.3% in the group that received dexamethasone among patients who needed mechanical ventilation.[12] A recent systemic overview on ARDS reported mortality rates since 2010: overall mortality rates of in-hospital: 45%, ICU: 38% and 28/30-day: 30%.[13]

### Rationale for the study and explanation of the hypothesis

Macrophages, part of the innate immune system, have become major therapeutic targets in ALI/ARDS. Macrophage activation is involved in the early phase of ARDS.[14] Alveolar macrophages (AMs) are the gatekeepers of the innate immune system in the lungs.

On activation, they secrete several inflammatory cytokines and chemokines including interleukin-1 beta (IL-1β), interleukin-6 (IL-6) and tumour necrosis factor alpha (TNF-α), to attract T-helper1 (Th1)/T-helper 17 (Th17)-cells, new macrophages and neutrophils. AMs are also responsible for clearing apoptotic neutrophils when the infection resolves. Proteomics involved in the switch from inflammatory macrophage (M1) to resolving or anti-inflammatory macrophage subtype (M2) was recently described in a human study of ALI/ARDS.[15] Hypoxia inducible factor-1 and factor-2 (HIF-1 and HIF-2) and inflammatory factors such as signal transducer and activator of transcription 3 (STAT3)

and nuclear factor kappa-light-chain-enhancer of activated b-cells (NFκB) are important transcription factors involved in macrophage polarisation. How and if it is possible to intervene with this intricate network of redox signalling is not clear.[16] Hyperbaric oxygen (HBO) has been used for almost a century, initially for decompression sickness (DCS), but it was soon noted that it had several anti-inflammatory effects.[17 18] Recent evidence from animal studies suggest that HBO ameliorate inflammation in DCS induced ALI through polarisation of macrophages from M1 to M2.[19 20] HBO has been shown to polarise macrophages from M1 to M2 associated with IL-10 and thereby reduces inflammation,[21 22] and 30-min HBO ex vivo inhibit monocyte IL-1β and TNF-α.[23] Patients presenting to hospital with COVID-19 normally have almost a week of mild or moderate flu-like symptoms but on admission often have an isolated hypoxic respiratory failure. Many patients, despite severe hypoxemia do not have dyspnoea or carbon dioxide retention suggesting a diffuse but moderate alveolar oedema and a hypoxic adaptation. Hypoxia is relative to the upregulation of adaptive mechanisms. When medical oxygen is administered for a prolonged period, the adaptive mechanisms are put out of play and might aggravate oxidative stress. HBO will give patients a short burst of oxidative stress and reactivate adaptive responses. The hypothesis of HBO as a safe and effective treatment and possible mechanisms has been previously published.[24–26] Published case series from China and the USA indicate that HBO in these patients may be safe and beneficial.[27–30] A propensity-matched control study (n=20) from the USA showed 50% lower mortality and almost two-thirds less need for mechanical ventilation in the HBO-treated group.[31]

HBO has the potential to reduce inflammation, restore normal defence mechanisms and thereby reduce morbidity and mortality in COVID-19 pneumonitis

### Remaining gap of evidence

HBO has been provided as 'compassionate use' for COVID-19 and some evidence from small case series and a prospective cohort suggests that it is safe and effective, but this needs to be confirmed in randomised controlled trials. There are concerns regarding oxygen toxicity in already inflamed lungs and the optimal dose and timing are still largely unknown. The multiple explanatory outcome measures in our trial may answer some of these questions. Here we report a summary of our protocol that adhere to International Council for Harmonisation-Good Clinical Practice (ICH-GCP) and Standard Protocol Items: Recommendations for Interventional Trials (SPIRIT) guidelines,[32] V.4 27 February 2021 of the protocol is available as online supplemental file 1 and substantial amendments will be available on clinicaltrials.gov or by request from the corresponding author. The SPIRIT checklist refers to the full protocol.

## Hypothesis and objectives

The overall hypothesis to be evaluated is that HBO reduce mortality, increase hypoxia tolerance and prevent organ failure in patients with COVID-19 pneumonitis by attenuating the inflammatory response.

The primary objective is to evaluate if HBO reduces the number of ICU admissions compared with best practice for COVID-19. Main secondary objectives are to evaluate if HBO reduces the load on ICU resources, morbidity and mortality in severe cases of COVID-19 and to evaluate if HBO mitigates the inflammatory reaction in COVID-19. Other secondary objectives (in selection) is to evaluate if HBO is safe for SARS-CoV-2 positive patients and staff.

## METHODS AND ANALYSIS

### Study design

Randomised, controlled, phase II, open label, multicentre trial conducted at hospitals with hyperbaric facility and ICU. The trial will investigate the safety and efficacy of HBO for COVID-19 but also multiple explanatory outcomes. The total number of participants will be 200 (100 per group) with a subgroup of 20 subjects for explanatory endpoints where we collect blood for extended immunology. Block randomisation will be performed, stratified by gender and site. The trial consists of nine visits over 30 days after randomisation, each visit consists of three parts; (a) review of medical records since last visit and documentation in the electronic case report form (eCRF), (b) measurements and actions to correct any deviations, (c) HBO treatment, if randomised (visit 1–7 only). A flowchart of the study design is depicted in figure 1. and The Consolidated Standards of Reporting Trials flow chart of the trial is depicted in figure 2.

### Setting and study subjects

The Sponsor is Karolinska Institute, Sweden and presently three centres in Sweden and Germany are involved.

Adult patients with SARS-CoV-2 infection, with at least two risk factors for increased mortality, likely to develop ARDS criteria and need intubation within 7 days of admission to hospital will be screened. After information and signed informed consent, study subjects will be checked for inclusion/exclusion criteria.

The inclusion/exclusion criteria are listed in table 1.

### Randomisation

Subjects will be enrolled and randomised consecutively as they are found to be eligible for inclusion in the study. HBO treatment will start within 24 hours of randomisation. Eligible subjects will be randomised in a 1:1 allocation, stratified by site and gender in blocks (blinded to all but the randomising clinical research associated at Karolinska Trial Alliance, KTA) to either HBO or Control. The randomisation sequence is computer-generated using RANDOMIZE.NET.

### Interventions

HBO in addition to best practice compared with best practice

HBO: HBO 1.6–2.4 Atmospheres Absolute for 30–60 min, maximum five treatments first 7 days

Control: best practice treatment for COVID-19

The first HBO treatment will be given within 24 hours after inclusion. Patients with respiratory symptoms admitted to the hospital will be informed and asked to participate. The patients will be included once they fulfil the inclusion criteria and none of the exclusion criteria, but the timing of the HBO treatment will depend on available resources.

### Measurements

After the patient has been informed about the study and if agreement to participate, an informed consent form (ICF) will be signed off before any study specific procedures occur.

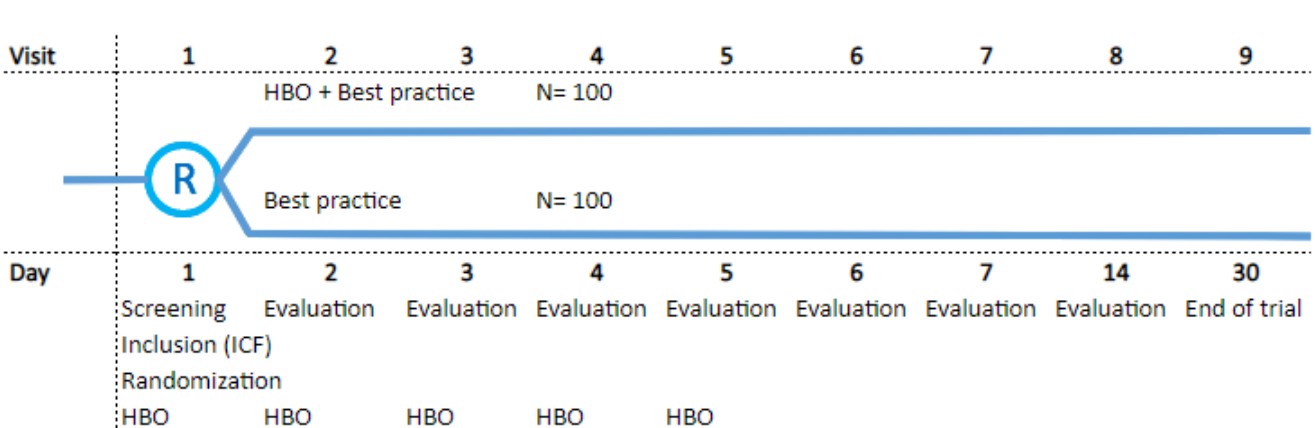

**Figure 1** Flowchart of the study design. HBO, hyperbaric oxygen; ICF, informed consent form.

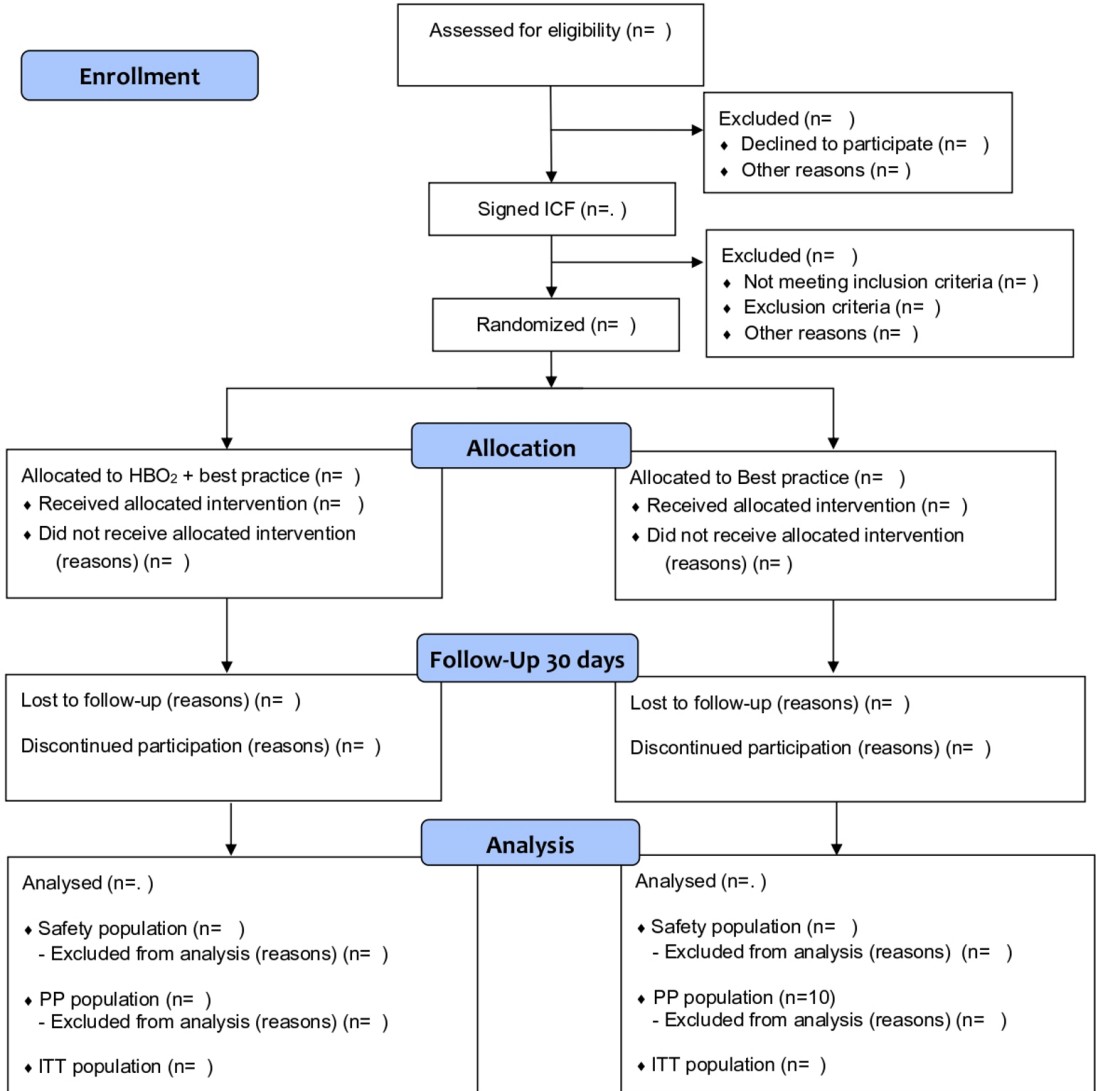

**Figure 2** The Consolidated Standards of Reporting Trials flow chart of the trial. HBO, hyperbaric oxygen; ICF, informed consent form; ITT, intent-to-treat; PP, per protocol.

During the screening, procedures to assure the patient's eligibility for the study participation will be performed. Females of childbearing potential will have a serum pregnancy test taken. Demographics, medical history including COVID-19 specific history and review of routine blood tests, secondary infections, viral load, radiology, concomitant medications before inclusion will be recorded. Mean new early warning score (NEWS) for the past 24 hours (three measurements 8, 14, 22±2 hours) will be recorded if available (mean is calculated after data are exported from eCRF at the end of study). Baseline NEWS at inclusion will also be recorded. A physical examination will be performed and a HBO-specific questionary as per local routine will be obtained. Subject will be randomised to either HBO (in addition to best practice) or best practice. Routine chemistry and study-specific blood tests will be collected. A complete list of procedures is listed in table 2.

### Trial endpoints
The primary endpoint is the proportion of subjects admitted to ICU from day 1 to day 30 based on predefined criteria for ICU admission. Main secondary efficacy endpoints are 30-day mortality, time to intubation, time to ICU admission and mean change in inflammatory response and main safety endpoints are measurement of adverse event (AE) and serious adverse events (SAE). A list of main efficacy and safety endpoints is listed in table 3.

### Safety and adverse events
An independent Data Safety Monitoring Board (DSMB) will evaluate the safety data in the context of the overall trial and the currently existing information about the study drug. The DSMB is composed of three experts in their respective disciplines of medicine, clinical trial methodology and conduct.

The DSMB will review the data during the course of the study, a charter delineating their guidelines for operating and stopping rules for terminating individual patients, a portion or all of the trial prematurely, was drawn up before the trial started. The members of the DSMB,

**Table 1** COVID-19-HBO: overview of inclusion and exclusion criteria

| Inclusion criteria | Aged 18–90 years |
| --- | --- |
| | $PaO_2/FiO_2$ (PFI) below 200 mm Hg (26.7 kPa) (based on ABG measurement) |
| | Suspected or verified SARS-CoV-2 infection |
| | At least two risk factors for increased morbidity/mortality<br>► Age above 50 years<br>► Hypertension<br>► Cardiovascular disease<br>► Diabetes or prediabetes<br>► Active or cured cancer<br>► Asthma/COPD<br>► Smoking<br>► D-Dimer >1.0 mg/L<br>► Autoimmune disease |
| Exclusion criteria | ARDS/pneumonia caused by other viral infections (positive for other virus) |
| | ARDS/pneumonia caused by other non-viral infections or trauma |
| | Known pregnancy or positive pregnancy test in women of childbearing age |
| | Patients with previous CT verified lung fibrosis more than 10% |
| | CT-verified or spirometry-verified severe COPD with emphysema |
| | Contraindication for HBO according to local guidelines* |
| | Not likely to need ICU admission within 7 days of screening (subjective criteria that may exclude any patients who fulfil the other inclusion criteria but where the treating physician suspect a spontaneous recovery) |
| | Mental inability, reluctance or language difficulties that result in difficulty understanding the meaning of study participation |
| | Prisoner |
| | Unable/risk to move patient to hyperbaric chamber |

*Contraindications are described in the Standard Operations Procedure for each centre; generally, includes claustrophobia, pneumothorax, severe COPD.
ARDS, acute respiratory distress syndrome; HBO, hyperbaric oxygen.

meeting plan and responsibilities are specified in the original protocol (p.8, 42–43).

The definition, handling, follow-up and reporting of AEs are defined in the original protocol (p.34–38).

## Statistical analysis

*Power calculation*: the primary endpoint ICU admission is defined by criteria for selection for ICU. We have assumed that 50% of the subjects will have at least one criterion during the course of the study and we aim to reduce the ICU admission rate by 40%, that is, to an ICU admission rate of 30%. To achieve 80% power with type-I error rate of 0.05 (two-tailed), a sample size of 93 subjects per group is required. We plan to enrol 200 subjects into this trial. Interim analyses may decide to recalculate sample-size for the trial.

Sample size calculation was done in nQuery V.7.

Primary and secondary endpoints will be evaluated using the intent-to-treat population (ie, all randomised subjects) and the primary endpoint also using the per protocol population (ie, all randomised subjects with no major protocol violations). All randomised subjects will be included in the safety population. The primary analysis of the primary endpoint will be performed using the Cochran Mantel Haenszel test adjusting for randomisation strata site and gender.

## Patient involvement

The study design and consent form were discussed with and approved by a patient representative. We thank Nanda Holm, patient contact at Rare diseases Sweden for her support.

## LIMITATIONS

The current trial has limitations and there are several potential threats to the validity and generalisability of the results. First, due to the nature of the epidemic, available resources, the risk of transport and contamination, it would be unethical and possibly unsafe to conduct a placebo-controlled trial. Second, 'Best practice' have changed over the course of the pandemic and it may differ between different countries and centres. In the evaluation of safety and efficacy these aspects will be considered. Third, the sample size calculation and risk factors are based on early pandemic data. The rationale for 1:1 randomisation is that this is a new disease and we will use a slightly lower dose than often used in more stable patients without ALI. Also, 1:1 allocation will maximise the statistical power. If the interim analysis can show supportive evidence for efficacy, the trial committee/ safety and data monitoring board may choose to change the randomisation to 2:1.

## ETHICS AND DISSEMINATION

HBO has the potential to prevent COVID-19 infection developing into ARDS and multiorgan failure and would then relieve ICU resources and potentially save lives. The nature of the disease with high mortality and no effective cure make the risk group a 'vulnerable group' and it is important to make sure that the subjects are not unduly influenced by the expectation or benefits associated with participation. Therefore, the study will be carried out in compliance with ICH-GCP, respective national legislation and according to the Declaration of Helsinki. The National Institutional review board in Sweden (Etikprövningsmyndigheten, Dnr: 2020-01705 Application date 27 March 2020 and approval date

**Table 2** COVID-19-HBO: list of procedures

| Activity | Visit 1 | Visit 2 | Visit 3 | Visit 4 | Visit 5 | Visit 6 | Visit 7 | Visit 8 | Visit 9 |
|---|---|---|---|---|---|---|---|---|---|
| Day | Day 1 | Day 2 | Day 3 | Day 4 | Day 5 | Day 6 | Day 7 | Day 14 | Day 30 |
| Screening | x | | | | | | | | |
| Inclusion/exclusion criteria | x | | | | | | | | |
| Pregnancy test if woman of childbearing age | x | | | | | | | | |
| HBO-specific medical history/physical examination | x | | | | | | | | |
| Signed informed consent form | x | | | | | | | | |
| Randomisation | x | | | | | | | | |
| 1. Medical history | x | | | | | | | | |
| 2. Demography* | x | x | x | x | x | x | x | x | x |
| 3. Concomitant medications | x | x | x | x | x | x | x | x | x |
| 4. NEWS score | x, x, x† | x, x, x | x, x, x | x, x, x | x, x, x | x, x, x | x, x, x | x, x, x | x, x, x |
| 5. Standard/study-specific biochemistry | x | x | x | x | x | x | x | x | x |
| 6. Study-specific CBG/ABG | x, x, x† | x, x, x | x, x, x | x, x, x | x, x, x | x, x, x | x, x, x | x, x, x | x, x, x |
| 7. Plasma (microRNA) | x | x | x | x | x | x | x | x | x |
| 8. CBG/ABG HBO | 3x | 3x | 3x | 3x | 3x | 3x | 3x | | |
| 9. HBO indicated/planned | x | x | x | x | x | x | x | | |
| 10. HBO treatment | x | x | x | x | x | x | x | | |
| 11. AE | x | x | x | x | x | x | x | x | x |
| 12. ADR | x | x | x | x | x | x | x | x | x |
| 13. UPTD | x | x | x | x | x | x | x | x | x |
| 14. CPTD | x | | | | | | | | x |
| 15. ICU admission | | x | x | x | x | x | x | x | x |
| 16. Intubation/mechanical ventilation | | x | x | x | x | x | x | x | x |
| 17. ICU mortality | | x | x | x | x | x | x | x | x |
| 18. Hospital mortality | | x | x | x | x | x | x | x | x |
| 19. Overall mortality | | x | x | x | x | x | x | x | x |
| 20. Secondary infections | x | x | x | x | x | x | x | x | x |
| 21. Viral load | x | x | x | x | x | x | x | x | x |
| 22. Staff safety (negative events) | x | x | x | x | x | x | x | x | x |
| 23. Pulmonary CT (check records) | x | x | x | x | x | x | x | x | x |
| 24. Chest X-ray (check records) | x | x | x | x | x | x | x | x | x |
| 25. Chest ultrasound (if available) | x | x | x | x | x | x | x | x | x |
| 26. Extended immunology (n=20) | x | | | x | | | x | x | x |

CBG/ABG HBO is collected once daily for the first 7 days and if clinically warranted.

All used acronyms and abbreviations are listed in the original protocol (p.9–10, Supplement).

Visit 1–7 is 8-7:59 and visits 8 and 9 are 7 days 8-7:59.

*Visits 2–9 demography check only involves change in DNR status.

†Depending on time of inclusion 1–3 samples/observations will be collected during visit 1 at the specified time points. Additionally, a baseline ABG (if not available from the patient's medical records) and a baseline NEWS is collected.

29 April 2020 (included a request for amendment 23 April 2020 and amended 23 April 2020). Approval by the Swedish Medical Product Agency (Läkemedelsverket) (LV: Application 23 April 2020 and decision 8 May 2020), Dnr 5.1-2020-36673. The trial was registered online prior to initiation on ClinicalTrials.gov (31 March 2020), NCT04327505 and on EU Clinical Trials Register (8 May 2020), EudraCT number: 2020-001349-37.

The trial is monitored by KTA, an independent organisation before the trial started, during the trial conduct and after the trial is completed, so as to ensure that the trial is carried out according to the protocol and that data are collected, documented and reported according to ICH-GCP and applicable ethical and regulatory requirements. Monitoring is performed as per the trial's monitoring plan and is intended to ensure that the subject's rights, safety and

**Table 3**  COVID-19-HBO: trial endpoints

| | |
|---|---|
| Primary endpoint | The proportion of subjects admitted to ICU from day 1 to day 30, based on at least one of the following criteria:<br>(1) Rapid progression over hours<br>(2) Lack of improvement on high flow oxygen >40 L/min or non-invasive ventilation with fraction of inspired oxygen ($FiO_2$) >0.6<br>(3) Evolving hypercapnia or increased work of breathing not responding to increased oxygen despite maximum standard of care available outside ICU<br>(4) Haemodynamic instability or multiorgan failure with maximum standard of care available outside ICU |
| Secondary endpoints | (in selection) |
| Main secondary efficacy endpoints | (1) Proportion of subjects with 30-day mortality, all-cause mortality, from day 1 to day 30.<br><br>(2) Time-to-intubation, that is, cumulative days free of invasive mechanical ventilation, from day 1 to day 30<br><br>(3) Time-to-ICU, that is, cumulative ICU free days, derived as the number of days from day 1 to ICU, where all ICU free subjects are censored at day 30<br><br>(4) Mean change in inflammatory response from day 1 to day 30<br>1. White cell count+differentiation<br>2. Procalcitonin<br>3. C-reactive protein<br>4. Cytokines (IL-6) (if available at local laboratory)<br>5. Ferritin<br>6. D-dimer<br>7. LDH<br><br>(5) Overall survival |
| Safety endpoints | (1) The number of subjects, proportion of subjects and number of events of AE<br><br>(2) The number of subjects, proportion of subjects and number of events of SAE<br><br>(3) The number of subjects, proportion of subjects and number of events of SADR<br><br>(4) Mean change in $PaO_2/FiO_2$ before and after HBO compared with mean variance in $PaO_2/FiO_2$ in the control group during day 1 to day 7<br><br>(5) Mean change in NEWS before and after HBO compared with mean change in daily NEWS in the control group during day 1 day 7<br><br>(6) Number of negative events in staff associated with treatment of subject (eg, contact with aerosol from subject), number of events from day 1 to day 30 or last day in hospital if subject is discharged earlier, or at withdrawal |

ABG, arterial blood gas; AE, adverse event; COPD, chronic obstructive pulmonary disease; HBO, hyperbaric oxygen; ICU, intensive care unit; LDH, lactate dehydrogenase; PFI, Ratio of arterial oxygen partial pressure to fraction of inspired oxygen; SADR, suspected adverse drug reaction; SAE, serious adverse events.

well-being are met as well as data in the eCRF are complete, correct and consistent with the source data. The monitoring will be performed by an independent experienced monitor qualified in ICH-GCP, applicable national and international regulations and the Declaration of Helsinki.

Results will be disseminated at national and international conferences and then published in international peer-reviewed scientific journals with open access. Positive, negative and any inconclusive results will be published.

## CURRENT TRIAL STATUS
The first site was initiated 20 May 2020, second site 29 November 2020, third site 14 June 2021. 31 subjects have been randomised. We conducted the first safety analysis 16/3 when 20 subjects had completed the trial and the first DSMB meeting was conducted 12/5 2021. The DSMB recommended to continue the trial as planned without modifications to the conduct/protocol. We are awaiting a forth wave and plan to initiate more centres during 2021.

**Author affiliations**
[1]Department of Physiology and Pharmacology, Karolinska Institutet, Stockholm, Sweden
[2]Perioperative Medicine and Intensive Care, Karolinska University Hospital, Stockholm, Sweden
[3]Department of Anaesthesia and Intensive Care, Blekinge Hospital Karlskrona, Karlskrona, Sweden
[4]Department of Anaesthesiology and Intensive Care Medicine, Catholic Charities Hospital, St. Josef, Regensburg, Germany
[5]Department of Anaesthesiology and Intensive Care Medicine, Bergmannsheil und Kinderklinik Buer GmbH, Gelsenkirchen, Germany
[6]Department of Anesthesiology and Intensive Care, University of Gothenburg Sahlgrenska Academy, Goteborg, Sweden
[7]Department of Molecular Medicine and Surgery, Karolinska Institutet, Stockholm, Sweden
[8]Department of Laboratory Medicine, Division of Clinical Microbiology, Karolinska Institute, Stockholm, Sweden
[9]Department of Mathematics and Natural Science, Blekinge Institute of Technology, Karlskrona, Sweden
[10]JK Biostatistics AB, Stockholm, Sweden
[11]Department of Radiology Solna, Karolinska University Hospital, Stockholm, Sweden

[12]Department of Mathematics and Natural Sciences, TIMN, Blekinge Institute of Technology, Karlskrona, Sweden

[13]SwAF Diving and Naval Medicine Centre, Swedish Armed Forces, Karlskrona, Sweden

[14]Department of Medicine Solna, Karolinska Institutet, Stockholm, Sweden

[15]Department of Respiratory Medicine and Allergy, Karolinska University Hospital, Stockholm, Sweden

[16]Center for Diabetes, Academic Specialist Center, Stockholm, Sweden

[17]Department of Reproductive Medicine, Division of Gynecology and Reproduction, Karolinska Universitetssjukhuset, Stockholm, Sweden

[18]Department of Oncology-Pathology, Karolinska Institutet, Stockholm, Sweden

[19]Department of Emergency Medicine, UCSD, La Jolla, California, USA

**Acknowledgements** We thank Georg Rinneberg, manager of the hyperbaric unit at Bergmannsheil und Kinderklinik Buer, Gelsenkirchen for his help with organising the trial in Germany. Clinical trial monitoring including conduct was done by Karolinska Trial Alliance, they also assisted with writing the protocol, eCRF, Laboratory manual, DSMB charter and IRB submission. Smart-Trial was used for creating the eCRF.

**Contributors** AK is the coordinating investigator who wrote the hypothesis and developed most of the protocol together with PL (sponsor representative). AK and PL wrote the applications to Swedish IRB and MPA. KAR-W, JD, JHK, MSi, PB, NO, SPN and OF contributed with information to the protocol and IRB/MPA applications. JD is principal investigator at Blekingesjukhuset. MTP is national coordinating investigator in Germany and principal investigator in Regensburg. MK is principal investigator in Gelsenkirchen. MK and MTP wrote the German IRB and MPA applications with assistance of AK. All authors (also including XZ, MSk and SC) contributed to the current submission and critically reviewed the manuscript. AK is corresponding author for this work, and attests that all listed authors meet authorship criteria and that no others meeting the criteria have been omitted.

**Funding** This work was supported by Vetenskapsrådet (KBF 2019–00446), made available by redirecting funds to COVID-19 research originally awarded to Kenny Rodriguez-Wallberg.

**Competing interests** Dr KAR-W reports grants from Vetenskapsrådet (Swedish Research Council), during the conduct of the study.

**Patient consent for publication** Not required.

**Provenance and peer review** Not commissioned; externally peer reviewed.

**ORCID iDs**
Anders Kjellberg http://orcid.org/0000-0002-4819-1024
Nicklas Oscarsson http://orcid.org/0000-0002-0460-1829
Xiaowei Zheng http://orcid.org/0000-0002-2648-1119
Peter Bergman http://orcid.org/0000-0003-3306-3713
Oskar Frånberg http://orcid.org/0000-0001-7051-3256
Mårten Silvanius http://orcid.org/0000-0002-4629-6324
Sergiu-Bogdan Catrina http://orcid.org/0000-0002-6914-3902
Kenny A Rodriguez-Wallberg http://orcid.org/0000-0003-4378-6181
Peter Lindholm http://orcid.org/0000-0002-0840-9244

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
