## [Reviewer comments · BMJ Open]

ARTICLE DETAILS

TITLE (PROVISIONAL)	A Randomized, Controlled, Open Label, Multicentre Clinical Trial to explore Safety and Efficacy of Hyperbaric Oxygen for preventing ICU admission, Morbidity and Mortality in Adult Patients With COVID-19
AUTHORS	Kjellberg, Anders; Douglas, Johan; Kraus, Michael; Pawlik, Michael; Oscarsson, Nicklas; Zheng, Xiaowei; Bergman, Peter; Frånberg, Oskar; Kowalski, Jan; Nyren, Sven; Silvanus, Mårten; Skold, Magnus; Catrina, Sergiu; Rodriguez-Wallberg, Kenny; Lindholm, Peter

VERSION 1 – REVIEW

REVIEWER	Grignola, Juan C. Universidad de la República Uruguay, Pathophysiology
REVIEW RETURNED	21-Dec-2020

GENERAL COMMENTS	I congratulate the authors for their exciting and well-proposed study protocol. Although there are several registered trials on the use of HBOT in patients with COVID-19, one of the main strengths of this trial is that it is a randomised controlled clinical trial. Patients hospitalized with COVID-19 have high mortality. As the authors stated, later reports showed 28-day mortality rates of 61.5% in ICU patients with acute respiratory illness. Furthermore, among the treatments that have proven to be effective, it would seem that only dexamethasone has been shown to reduce mortality significantly, highlighting the role of the activation of the innate inflammatory pathway involving the inflammasome activation. Hyperbaric oxygen therapy (HBOT) by definition means treatment with 100% oxygen at higher than atmospheric pressure, attenuating the production of pro-inflammatory cytokines. Therefore, HBOT could be used in COVID-19 to reduce cytokine levels, hampering catastrophic inflammation development. The present study protocol proposes to evaluate if HBOT reduces the number of ICU admissions compared to Best practice for COVID-19 prospectively. They also propose if HBOT reduces the load on ICU resources, morbidity, and mortality in the severe case of COVID-19. Additionally, the authors will explore the possible explanatory mechanisms that could be involved deriving from the administration of HBOT. I have some methodological points that should be clarified and some items to be added to improve the study: Introduction: I recommend adding a review about the role of HBOT for COVID-19 published very recently: Adv Exp Med Biol 2020:
---

	10.1007/5584_2020_568. Paganini M et al. It will enrich the study's rationale section (i.e., Inflammasome) Page 4, line 135. Methods and Analysis: On page 6, line 194: it is advisable to refine the selected risk factors for increased morbidity/mortality. Multiple risk scores predictive models have developed for COVID-19 to date, being designated to estimate the risk at different times of the disease (in the community, in-hospital mortality score -4C mortality score). While requiring further external validation, I recommend adding some of the following risk scores, particularly the CALL score, which predicts the progression risk in patients with COVID-19 (CID, 2020; 71:1393-99) and a novel COVID-19 30-day mortality score (Sci Rep 2020; 10:21379) to improve the evaluation of the impact of the HBOT. In inclusion criteria (table 1): add the units of D-dimer In exclusion criteria (table 1), the authors state that "Contraindication for HBO according to local guidelines" would be advisable to specify what the contraindications are. On page 8, Interventions: what is the argument of the proposed HBOT protocol? How will the HBOT be applied? What type of hyperbaric chamber will be used? There are side effects associated with HBOT. On page 11, the last paragraph, the authors state that adverse events are defined in the original protocol that we could not access. It is essential to identify and quantify the HBOT side effects for prevention, management, and informed consent. Given the possibility of pulmonary barotrauma (application of the hyperbaric therapy on an open-air space system like the lungs) and pulmonary oxygen toxicity (hyperoxia therapy), what type of care will be taken into account the presence of severe pulmonary disease?
--	---

REVIEWER	Hemmat, Nima Tabriz University of Medical Sciences, Immunology Research Center
REVIEW RETURNED	25-Dec-2020

GENERAL COMMENTS	This manuscript was designated to make a promising therapeutic approach in COVID-19 treatment done by Kjellberg et al. and in my side it has sufficient quality for publication.
--

REVIEWER	Gorenstein, Scott NYU Long Island School of Medicine
REVIEW RETURNED	27-Dec-2020

GENERAL COMMENTS	The study is well designed but until there is a significant number of patients enrolled there are no conclusions that can be drawn. Maybe consider re-submitting when 50% enrollment is achieved as an interm analysis.
---

VERSION 1 – AUTHOR RESPONSE

Reviewer: 1
Dr. Juan C. Grignola, Universidad de la República Uruguay
Comments to the Author:

I congratulate the authors for their exciting and well-proposed study protocol. Although there are several registered trials on the use of HBOT in patients with COVID-19, one of the main strengths of this trial is that it is a randomised controlled clinical trial.

Patients hospitalized with COVID-19 have high mortality. As the authors stated, later reports showed 28-day mortality rates of 61.5% in ICU patients with acute respiratory illness. Furthermore, among the treatments that have proven to be effective, it would seem that only dexamethasone has been shown to reduce mortality significantly, highlighting the role of the activation of the innate inflammatory pathway involving the inflammasome activation. Hyperbaric oxygen therapy (HBOT) by definition means treatment with 100% oxygen at higher than atmospheric pressure, attenuating the production of pro-inflammatory cytokines. Therefore, HBOT could be used in COVID-19 to reduce cytokine levels, hampering catastrophic inflammation development.

The present study protocol proposes to evaluate if HBOT reduces the number of ICU admissions compared to Best practice for COVID-19 prospectively. They also propose if HBOT reduces the load on ICU resources, morbidity, and mortality in the severe case of COVID-19. Additionally, the authors will explore the possible explanatory mechanisms that could be involved deriving from the administration of HBOT.

I have some methodological points that should be clarified and some items to be added to improve the study:

Thank you for a thorough and very constructive review, please read our response below:

Introduction:

I recommend adding a review about the role of HBOT for COVID-19 published very recently: Adv Exp Med Biol 2020; 10.1007/5584_2020_568. Paganini M et al. It will enrich the study's rationale section (i.e., Inflammasome) Page 4, line 135.

Thank you for the suggestion we have now added the reference, in the section with the other hypotheses on Page 5, line 151. It is also included in the full protocol v.4 (supplementary material)

Methods and Analysis:

On page 6, line 194: it is advisable to refine the selected risk factors for increased morbidity/mortality. Multiple risk scores predictive models have developed for COVID-19 to date, being designated to estimate the risk at different times of the disease (in the community, in-hospital mortality score -4C mortality score). While requiring further external validation, I recommend adding some of the following risk scores, particularly the CALL score, which predicts the progression risk in patients with COVID-19 (CID, 2020; 71:1393-99) and a novel COVID-19 30-day mortality score (Sci Rep 2020; 10:21379) to improve the evaluation of the impact of the HBOT.

This is a valid point that we have discussed. The full protocol was submitted to the IEC and IRB already in March 2020 and the trial was initiated in May 2020. We know much more today regarding risk factor but as you state risk scores needs to be further validated. Since this trial is conducted in accordance with ICH-GCP a change of inclusion criteria would be a substantial amendment that needs to be approved by the IRB in Sweden and Germany; therefore, do not wish to change the inclusion criteria, at least not until we have conducted the first interim analysis. We will take this into account if we submit a substantial amendment to the full protocol. To clarify this, we have added this information to "limitations", page 12, line 298.

In inclusion criteria (table 1): add the units of D-dimer

We have added the unit mg/L

In exclusion criteria (table 1), the authors state that "Contraindication for HBO according to local guidelines" would be advisable to specify what the contraindications are.

Contraindications differ slightly between centers and are individually assessed regarding to circumstances and type of chamber used, these are described in detail in the Standard Operating Procedure (SOP), in each Investigator Site File (ISF) but we have added some examples; page 7, line

204-205.

On page 8, Interventions: what is the argument of the proposed HBOT protocol? How will the HBOT be applied? What type of hyperbaric chamber will be used?

This is described in detail in the SOP and in summary in the full protocol (supplementary material), different centers will use different type of chamber and slightly different protocols.

There are side effects associated with HBOT. On page 11, the last paragraph, the authors state that adverse events are defined in the original protocol that we could not access. It is essential to identify and quantify the HBOT side effects for prevention, management, and informed consent. Given the possibility of pulmonary barotrauma (application of the hyperbaric therapy on an open-air space system like the lungs) and pulmonary oxygen toxicity (hyperoxia therapy), what type of care will be taken into account the presence of severe pulmonary disease?

We regret that you did not have access to the full protocol, this was submitted as supplementary material to the editor. We hope that this will be made available to you for a second revision and it will be made available in a future publication. We discuss this in detail in the risk/benefit evaluation section 3 in the full protocol. Severe COPD is one of the contraindications in the trial.

Reviewer: 2

Dr. Nima Hemmat, Tabriz University of Medical Sciences

Comments to the Author:

This manuscript was designated to make a promising therapeutic approach in COVID-19 treatment done by Kjellberg et al. and in my side it has sufficient quality for publication.

Thank you for your support.

Reviewer: 3

Dr. Scott Gorenstein, NYU Long Island School of Medicine

Comments to the Author:

The study is well designed but until there is a significant number of patients enrolled there are no conclusions that can be drawn. Maybe consider re-submitting when 50% enrollment is achieved as an interim analysis.

Thank you for your comments. The aim of this manuscript to BMJ is to publish the protocol prior to any data analysis. We have received some valuable peer-review feedback for future amendments to the protocol that may help us with future publication of the results. It may also aid others in designing similar trials.

Reviewer: 1

Competing interests of Reviewer: None declared.

Reviewer: 2

Competing interests of Reviewer: None declared

Reviewer: 3

Competing interests of Reviewer: Principal Investigator in another HBO - COVID trial

VERSION 2 – REVIEW

REVIEWER	Grignola, Juan C. Universidad de la República Uruguay, Pathophysiology
REVIEW RETURNED	17-Mar-2021
GENERAL COMMENTS	The authors have addressed the questions raised.